# Gradual Drug Release Membranes and Films Used for the Treatment of Periodontal Disease

**DOI:** 10.3390/membranes12090895

**Published:** 2022-09-17

**Authors:** Nausica Petrescu, Bogdan Crisan, Ovidiu Aghiorghiesei, Codruta Sarosi, Ioana Codruta Mirica, Ondine Lucaciu, Simina Angela Lăcrimioara Iușan, Noemi Dirzu, Dragos Apostu

**Affiliations:** 1Department of Oral Health, Iuliu Hatieganu University of Medicine and Pharmacy, 400012 Cluj-Napoca, Romania; 2Department of Maxillofacial Surgery and Oral Implantology, Iuliu Hatieganu University of Medicine and Pharmacy, 400029 Cluj-Napoca, Romania; 3Institute of Chemistry Raluca Ripan, Department of Polymer Composites, Babes-Bolyai University, 400294 Cluj-Napoca, Romania; 4Medfuture Research Center for Advanced Medicine, School of Medicine, Iuliu Hatieganu University of Medicine and Pharmacy, 400347 Cluj-Napoca, Romania; 5Department of Orthopaedics and Traumatology, Iuliu Hatieganu University of Medicine and Pharmacy, 400012 Cluj-Napoca, Romania

**Keywords:** periodontitis, drug, film, membrane, gradual drug release

## Abstract

Periodontitis is an inflammatory disease that, if not treated, can cause a lot of harm to the oral cavity, to the patients’ quality of life, and to the entire community. There is no predictable standardized treatment for periodontitis, but there have been many attempts, using antibiotics, tissue regeneration techniques, dental scaling, or root planning. Due to the limits of the above-mentioned treatment, the future seems to be local drug delivery systems, which could gradually release antibiotics and tissue regeneration inducers at the same time. Local gradual release of antibiotics proved to be more efficient than systemic administration. In this review, we have made a literature search to identify the articles related to this topic and to find out which carriers have been tested for drug release as an adjuvant in the treatment of periodontitis. Considering the inclusion and exclusion criteria, 12 articles were chosen to be part of this review. The selected articles indicated that the drug-releasing carriers in periodontitis treatment were membranes and films fabricated from different types of materials and through various methods. Some of the drugs released by the films and membranes in the selected articles include doxycycline, tetracycline, metronidazole, levofloxacin, and minocycline, all used with good outcome regarding their bactericide effect; BMP-2, Zinc–hydroxyapatite nanoparticles with regenerative effect. The conclusion derived from the selected studies was that gradual drug release in the periodontal pockets is a promising strategy as an adjuvant for the treatment of periodontal disease.

## 1. Introduction

Periodontitis (P-D) is a chronic multifactorial inflammatory disease associated with a dysbiotic biofilm and characterized by progressive destruction of the complex tooth-supporting apparatus (consisting of bone, ligaments, cementum, and surrounding connective tissues), which can lead to tooth loss. While the etiology of this chronic inflammatory disease is multifactorial, subgingival bacteria organized in biofilms are accepted as the primary etiological factor [1,2,3,4]. P-D is the sixth most prevalent condition in the world [5], affecting, according to the World Health Organization, around 10.8% of adults [6], representing a major public health issue worldwide with multiple scientific, social, economic, and cultural implications. The socio-economic and cultural point of view is linked to the impact of infection, multiple treatments, and edentations on patients’ health and quality of life [7]. Due to its high prevalence, in Europe, in 2018 P-D caused a EUR 149.52 billion loss [8] with over 743 million persons affected, and P-D is responsible for 3.5 million years of living with disability [9]. According to estimates, severe P-D alone costs the world USD 54 billion annually in lost productivity, and P-D overall has a significant economic impact, accounting for a large portion of the USD 442 billion in direct and indirect costs of oral disorders in 2010 [10]. These results show the huge social and economic burden of P-D. Socio-economic gradients in the rates of P-D have been reported in low-income countries [6] and disproportionally affect socially disadvantaged people. Social life and self-esteem are strongly linked to facial appearance, associated with the presence of teeth in the oral cavity. P-D is the leading cause of tooth loss in the adult population around the world, putting these people at risk for multiple tooth losses, edentation, and masticatory dysfunction, which can have a negative impact on their nutrition, quality of life, and self-esteem. P-D can also affect the occurrence and progression of various systemic diseases [11], even leading to overall death [12] in addition to having significant socioeconomic effects and medical expense implications. The importance of this issue relies on the need for new knowledge and the development of new products that would ensure regeneration of all affected periodontal structures (alveolar bone, cementum, periodontal ligaments) while delivering local antibiotherapy representing a highly efficient, low-cost treatment in P-D.

Nowadays the current treatment options in the field of P-D treatment consist of: open flap debridement (OFD) (mechanically disrupting and reducing the subgingival bacterial biofilm) [13], and systemic antibiotherapy, associated with the use of guided tissue regeneration strategies [14]. The limitations of the current approaches are proved by their limited long-term outcomes, with frequent relapses, thus needing periodic curettage and long term systemic antibiotherapy.

A new generation of nanostructured barrier membranes (NM) with enhanced properties is under development, because of their ability to increase cell adhesion, migration, proliferation and cell differentiation, thus promoting regenerative outcomes [15,16,17] and delivering local antibiotherapy. The ideal therapy with predictable long-term results, combining local targeted antibiotherapy with local regenerative strategies in a resorbable NM, is still debated in the international scientific literature and is not available on the market. The difficult element of the issue consists in ensuring both the regeneration of all affected periodontal structures (alveolar bone, cementum, periodontal ligaments) and delivering local antibiotherapy.

Systemic drug administration has limited effect, due to small local drug concentration [18] and the occurrence of antibacterial resistance [19], and is associated with systemic side-effects such as drug-induced hepatitis, nephrotoxicity, or myelosuppression [18]. The local administration of antibiotics is considered to be a promising treatment strategy, but more research needs to be done to establish a protocol and its long-term predictable success [20]. Among other drugs, minocycline [21], metronidazole, amoxicillin [22], and doxycycline [23] have proven to be efficient against periodontal pathogens. The local delivery of antibiotics with gradual and prolonged release has been highly investigated in the pursuit to find the most suitable drug delivery device [24,25].

Different regeneration inducers such as BMP-2 and Zinc–hydroxyapatite nanoparticles have been used in an attempt to restore affected periodontal tissue.

The characteristics of the carrier must be known in order to choose it for drug delivery in P-D, including the following: biocompatibility; if it is degradable or not; its ability to entrap and gradually release the drug; its stability; its ability to incorporate hydrophilic of hydrophobic substances; physical properties that make it easy to be inserted into the periodontal pocket, etc. [26]. The carriers can take various shapes or forms, as follows: strips, films, chips, membranes [26]. Each of those can be developed from various types of materials.

Films are generally developed from a polymeric matrix which can entrap the drug. Regarding their composition, various types of materials have been used for film production and tested for their drug delivering abilities: polymers (ethyl methacrylate-chlorotrimethyl ammonium methyl methacrylate) used for delivering clindamycin [25], cross-linked fish gelatin with chlorhexidine hydrochloride or chlorhexidine diacetate [27], atelocollagen preparations with tetracycline, etc. The film usually consists of a mixture of polymers, additives, drugs, and a solvent which can be hydrophilic or hydrophobic. Films can be degradable or nondegradable. Degradable films are preferred for their feasibility when it comes to their usage in the dental office: they do not have to be removed, so the patient does not have to return to the office and, because of their degradability, will not generate a foreign body response [27].

Considerable attention has been drawn to membranes, particularly electrospun membranes. Electrospinning is an easy, cost-effective method for fabricating fibers of different sizes [28]. The resulting membrane can imitate an extracellular matrix, which enhances cell migration, adhesion, and proliferation [29] and they have a high porosity, which leads to nutrient and oxygen diffusion [30]. These membranes can incorporate several drugs at the same time and release those drugs progressively to promote tissue regeneration or to have an antibacterial effect. Their main benefits are that they reduce the number of the patients’ dental appointments, and they also permit minimization of the drug dosage to avoid side effects [31].

The aim of this review was to assess medications loaded on various membranes and films and their impact and outcome on periodontal disease.

## 2. Materials and Methods

### 2.1. Search Strategy

A comprehensive literature search was conducted in PubMed, to find the answer to the following query: “Which drugs loaded on different membranes and films can induce periodontal regeneration?” Various term combinations were used in the search technique throughout the past 10 years, including “periodontitis, membrane, drug” “periodontitis, film, drug” and “periodontitis, film, antibiotic.” The database literature search was carried out separately by four academics, and the results were then compared. Divergences in the evaluation were addressed by conversation and re-evaluation of the article with the fifth researcher.

### 2.2. Inclusion and Exclusion Criteria

Articles published between 15 May 2012 and 15 May 2022 in the databases specified above met the inclusion requirements. We included research that involved using antimicrobial or bone-inducing membranes for periodontal regeneration in both animal and human studies.

Among the exclusion criteria, there were: articles with no available full text; articles published in a language other than English; systematic reviews or meta-analyses; case reports; experimental or in vitro studies; medical hypotheses; and articles discussing not loaded membranes.

### 2.3. Study Selection and Data Collection

The first step was to determine whether the titles and abstracts of the chosen papers were pertinent to the requested research. Based on the inclusion and exclusion criteria, a thorough text analysis of each eligible article was then carried out. All publications that satisfied the eligibility requirements were chosen, and data on the authors, year of publication, study design, subjects and study sample, type of membrane, loaded membrane, and results were collected using a standardized table. Initial selections of 638 articles from the database were made using the aforementioned search criteria. Following the removal of duplicate articles (42), 596 articles were left, which were then thoroughly examined based on inclusion and exclusion criteria, full-text accessibility, and type of article. A total of 12 papers were chosen to be included in this review because they met the qualifying requirements (Figure 1).

Using the Newcastle–Ottawa Scale, the quality of the studies included in this review was evaluated (NOS) [32]. Three quality parameters are quantified by the NOS scale (selection, comparability and outcome). There are eight distinct categories created from these parameters. Except for comparability, which can receive up to two points, each item on the scale is given a maximum score of one. Each given point was marked with a star in the table below. The highest possible score for each study is 9 [33]. Studies with NOS values between 0 and 3 were rated as low quality, 4–6 as moderate quality, and 7–9 as high quality (Table 1).

## 3. Results

Figure 1 shows how the papers for this literature review were located and chosen in accordance with PRISMA guidelines. A total of 638 articles from PubMed initially matched the search terms, of which 42 duplicates were found and eliminated. The remaining 596 articles were evaluated; 48 were eliminated because they did not discuss loaded membranes; 14 discussed the use of membrane in bone reconstruction; 3 were not fully accessible; 9 articles were literature reviews; 501 articles were in vitro studies; 7 case reports; 1 was a medical hypothesis; and 1 was a recommendation. Ultimately, 12 studies that described the usage of membranes loaded with antibacterial and regenerative drugs were included in this study following a thorough investigation (Figure 1).

The selected 12 articles were assessed and are presented in two tables, one addressing studies on humans and one addressing studies on animals. Data collected from the 12 articles were introduced in the two tables, regarding the following criteria: references (authors, year of publication), type of subjects/number, type of membrane, type of load, and obtained results (Table 2 and Table 3).

Four articles described the use and outcome of drug loaded membranes on humans.

Local delivery of doxycycline was tested by Mahajania et al. [34] in the treatment of the Gram-negative bacterial plaque found in periodontitis. The drug delivery system was obtained from hydroxypropyl methylcellulose powder and doxycycline powder, which were centrifuged and kept at 300° for three days in order to obtain 0.15 mm-thick sheets. The study was performed on 19 patients, at the level of the mandibular first and second molar, where the authors performed radiographic and clinical examinations, as well as microbial sampling for a period of 10 weeks. The drug delivery system consisting of doxycycline hydroxypropyl methylcellulose membranes was applied in the periodontal pocket on one mandibular molar site, while a control site on the other quadrant mandibular molar received a resorbable membrane containing hydroxypropyl methylcellulose membranes. The gingival index scores were superior in the doxycycline group and the patients were declared clinically healthy at the end of the study in the treatment group. No important differences were observed between groups during the probing depth measurements, while the treatment groups showed an important reduction of the bacterial count. Overall, doxycycline hydroxypropyl methylcellulose membranes have generated statistically significant improvements in clinical and microbial parameters in periodontal disease.

In a clinical prospective study on 30 systemically healthy non-smoking individuals undergoing periodontal therapy, EDTA root surface etching was examined by Gamal and Kumper (2012) [35] as a potential indirect delivery method to improve the availability of doxycycline (DOX) in the gingival crevicular fluid (GCF) following its release from the collagen membrane. Inclusion criteria were: contralateral interproximal intrabony defects (IBC) ≥ 4 mm; interproximal periodontal depth (PD) ≥ 6 mm; and clinical attachment level (CAL) ≥ 4 mm after phase one therapy of scaling and root planning (SRP). After phase one of periodontal treatment, subjects were divided into two groups. Open-flap debridement (OFD) was performed in both groups. Group 1 (G1) sites received treatment with the placement of DOX gel-loaded collagen (DOX-COL), whereas Group 2 (G2) sites received treatment with the implantation of DOX-COL following EDTA etching of the exposed root surfaces (DOX-COL + EDTA). Six months after treatment, significant clinical improvements were obtained in PD reduction and clinical attachment gain was observed in DOX–COL + EDTA- and DOX–COL-treated sites (*p* ≤ 0.05). Sites treated with DOX-COL + EDTA showed more pronounced clinical improvements than DOX-COL-treated sites. The reduction of the IBC depths in both groups was also reported, with a significant difference between G1 when compared with that of G2 in favor of the DOX–COL + EDTA group. The clinical 6-month follow-up data indicated that significant improvements were obtained for deep IBC after EDTA root surface etching and subsequent placement of DOX gel-loaded COL when compared with non-etched DOX–COL-treated sites.

The results of the study done by Gamal and Kumper 2012 [35] suggest that the DOX–COL + EDTA guided tissue antibacterial regimen is a convenient method of obtaining a prolonged drug release without compromising the space that should be occupied by regenerating tissues or by inducing smearing of the root surface by DOX gel.

According to Khan et al., 2015 [36], a localized controlled delivery approach is preferred to achieve high local bioactivity and low systemic side effects of antibiotics in the treatment of periodontal infections. They created chitosan (CS) films containing metronidazole (MZ) and levofloxacin (LF) for their investigation in an effort to maintain regulated medication concentrations above the minimum inhibitory concentration (MIC) for a protracted period of time. In this work, the films were made using a simple solvent-casting method without the use of any potentially dangerous organic solvents. A total of 10 patients (20–50 years) screened for signs of chronic periodontitis, presenting four sites of periodontal affected teeth were included in this research. The 40 periodontal sites were divided into four groups: group 1–scaling and root planning (SRP); group 2–SRP + placebo film; group 3–SRP with a film containing LF; group 4–SRP with a film containing LF and MZ. Although the standard approach, SRP, used to treat periodontal pockets is beneficial, group 4 patients who had SRP in addition to MZ and LF demonstrated greater improvement than group 1 patients who received SRP alone. After 8 weeks of treatment, the gum had lightened to a pale pink color with no symptoms of swelling or inflammation. The periodontal ligament had also quickly reattached, and the depth of the pocket had significantly decreased. MZ + LF films were therefore superior to LF films or SRP alone in terms of managing periodontal health.

The objective of Kassem et al. 2014 [37], was to develop two types of polyelectrolyte complex (PEC) films using chitosan-alginate and chitosan-pectin and loaded with Tetracycline hydrochloride (Tc), in order to be used for the local treatment of periodontitis. After the films were prepared, loaded and tested in vitro, they were applied in the periodontal pockets of five human patients. After the films were inserted into the periodontal pockets, they were each covered with a periodontal pack. The periodontal pockets were measured before and after film application. The films were removed after 7 days and the follow-up was continued for 14 days. After film removal, the gums showed no inflammation signs, no bleeding on probing and the depth of the periodontal pockets was significantly reduced. Tetracycline hydrochloride (Tc)-loaded chitosan–alginate and chitosan–pectin PEC films could be used in the treatment of periodontal pockets.

Liu et al., 2020 [38] developed a micelle nanofiber membrane by coaxial electrospinning, and further enhanced with SP600125 in the shell and BMP-2 in the core. After proper testing of the release behavior and osteoinductive capacity, the animal test was performed on nine beagle dogs. Class II furcation lesions of 5 mm diameter and 2 mm in depth were created at the level of the second, third and fourth mandibular teeth. The membranes were divided into -/- (no treatment), SP600125/-, -/BMP-2 and SP600125/BMP-2. The treatment membranes were implanted 4 weeks after the first procedure. Moreover, a control group with sham surgeries were included in the study. Micro-CT and histological examinations were performed. Histological examinations did not show statistically significant differences between the groups at two months. The bone volumes using micro-CT examinations were statistically significantly higher in the SP600125/BMP-2 compared to SP600125/- and -/BMP-2.

Khajuria et al., 2017 [40], in an experimental study, employed the Porphyromonas gingivalis–lipopolysaccharide injections (Pg-LPS)-induced periodontitis in a rat model to mimic the clinical traits of periodontitis-like inflammation and loss of alveolar bone. Chitosan-based risedronate/zinc-hydroxyapatite intrapocket dental film (CRZHDF) might play a role in the effective limitation of periodontal inflammation and alveolar bone loss due to the complementary mechanisms of action of risedronate, zinc-hydroxyapatite (zinc-HA), and chitosan. The fabricated CRZHDF-A (0.1% *w/v* risedronate and 0.1% *w/v* zinc-HA nanoparticles) and CRZHDF-B (0.2% *w/v* risedronate and 0.2% *w/v* zinc-HA nanoparticles) were subdivided into smaller rectangular inserts (2 mm × 1 mm) by punching out. The authors assessed the effects of CRZHDF in the treatment of periodontitis on a rat model (12 subjects), divided into five groups: the healthy group; the untreated periodontitis group; the periodontitis plus CRZHDF-A group; the periodontitis plus CRZHDF-B group; and the plain chitosan group. The results of this study show that local administration of CRZHDF-A and CRZHDF-B induced remarkable inhibition of alveolar bone loss. In contrast, the mean values of alveolar bone resorption were statistically lower in the CRZHDF-A, CRZHDF-B and plain chitosan groups with respect to the untreated periodontitis group. Significant decrease in the mean values of alveolar bone resorption was observed in CRZHDF-A and CRZHDF-B groups in comparison to the plain chitosan group. This study affirms that novel CRZHDF treatment significantly inhibits regional alveolar bone destruction and contributes to periodontal healing in an experimental periodontitis rat model.

Due to the complementary mechanisms of action of chitosan and metformin, Khajuria et al. (2018) [39] studied the effect of chitosan-metformin based intrapocket dental film (CMIDF) in reducing periodontal inflammation as well as alveolar bone destruction in periodontitis. They assessed the effect of CMIDF on Pg-LPS coupled ligature-induced experimental periodontitis in rats to describe the pharmacological effect of CMIDF on alveolar bone degradation in periodontitis. Additionally, in vitro tests using specific strains of Tannerella forsythia and Porphyromonas gingivalis examined the antibacterial effectiveness of CMIDF. A casting method was employed to create chitosan inserts that were loaded with metformin. For this study, the authors prepared CMIDF-A and CMIDF-B films with two different ratios of metformin and chitosan i.e., 1:75 and 1:125, respectively. Five groups of eight rats were used: the control group; the untreated periodontitis group; the periodontitis plus CMIDF-A group; the periodontitis plus CMIDF-B group; and the periodontitis plus plain chitosan film group. By preventing the development of periodontal pathogens including *P. gingivalis* and T. forsythia in vitro, the produced intrapocket dental films demonstrated good antibacterial efficacy. Additionally, this research provided additional evidence that local administration of CMIDF into the periodontal pockets of rats had the ability to reverse the loss of alveolar bone. This research shows potential antibacterial and osteoprotective efficacy of novel CMIDF in experimental periodontitis and needs further evaluation of CMIDF in patients suffering from periodontitis.

The aim of Ma et al., 2020 [41] in their study was to develop a protocol for a poly (lactic-co-glycolic acid) electrospun membrane (PLGA) loaded with Minocycline (MINO-PLGA). After MINO-PLGA was tested in vitro, it was tested on animal model—25 rats divided into five experimental groups: one control group with no procedure done and no ligature applied; one group with ligature on the first molar to induce periodontal disease; one group with ligature and PLGA; one group with ligature and MINO-PLGA; and one group with ligature and Periocline (hydroxyethyl-cellulose matrix containing minocycline hydrochloride). The ligatures were orthodontic steel wires and were maintained for 4 weeks. After 4 weeks, the ligatures were removed and replaced with nothing (control group) or with one of the treatments described above for each experimental group. The histological analyses showed an increase of the alveolar bone height for the last two experimental groups: MINO-PLGA and Periocline and the periodontal tissues were healthier at 3 and 6 weeks, respectively. The same experimental groups at the same periods of time revealed a higher alveolar crest, compared with the other groups and the MINO-PLGA group had significantly better results at 6 weeks, compared with Periocline group.

The study published by Ho et al. in 2021 [42] had the purpose to enhance periodontal regeneration using an antibiotic-loaded membrane. The poly-DL-lactic acid membrane loaded with Amoxicilin (PDLLA-AMX) was first tested in vitro and, after, in vivo on 27 Sprague Dawley rats. Silk ligatures were placed on the second molars on both sides of the maxilla, in order to have 54 examined sites where they induced periodontal disease. After 7 days, the ligatures were removed and three types of membranes were inserted into the periodontal pouch: collagen membranes with and without nanofibers and collagen membrane loaded with Amoxicilin nanofibers. The histological and Micro-CT analysis revealed lower inflammation and earlier periodontal tissues regeneration for the PDLLA-AMX group.

Wu et al. 2021 [43] used, for the treatment of periodontitis, hydrogel membrane homogeneously incorporated with ZnO nanoparticles (ChT-1%ZnO). The authors demonstrated the effect of the membrane on a Wistar rat model. Subjects were divided into four groups: (1) No-defect, (2) No-membrane, (3) ChT, (4) ChT-1%ZnO. A periodontal defect of 1.5 mm length, 1 mm depth, and 1 mm width was performed on the mesial side of the first upper molar. In group 3 and 4, ChT and ChT-1%ZnO membrane were used to cover the defects. MicroCT and histological analysis was performed to evaluate the outcomes. The results suggested that ChT-1%ZnO performs better than pure ChT in periodontal bone defects regeneration, but it could not fully repair the defect to the primary level (1.032 mm). The membrane could efficiently facilitate the repair when compared with the No-membrane group.

Li et al. [44] proposed a local drug delivery system (LDDS) as a monotherapy or an adjunct to conventional therapy for periodontitis. For this purpose, they developed an enzyme-mediated periodontal membrane for targeted antibiotic delivery into infectious periodontal pockets, using a chitosan membrane containing polyphosphoester and minocycline hydrochloride (PPEM) prepared by the casting method. First, the membrane was evaluated in vitro. For in vivo testing of the PPEM membrane, 12 male Sprague–Dawley rats were used, assigned randomly to three groups (control group, periodontitis group and periodontitis with PPEM treatment group). A ligature wire was placed into the gingival sulcus and tied around the maxillary second molar to induce periodontitis and bone loss. For PPEM treatment group, after SRP the membrane was placed in the periodontal pocket and adhesive periodontal dressing was used to protect the wounds and membranes for 2 weeks. Maxillary samples were scanned by a microCT scanner, and 3D images were acquired. Gingival index (GI) was evaluated and classified according to the degree of gingival inflammation. Six anatomical sites in two maxillary molars were measured for the distance from the alveolar bone crest to the cementoenamel junction, which represents the level of alveolar bone loss. The periodontitis group showed much higher GI and alveolar bone loss (ABL), compared to the control group. After 4 weeks of treatment, GI and ABL in the PPEM group were significantly lower, compared to the untreated periodontitis group, showing that the incorporation of 0.6% (*w*/*v*) MH drug provided the membrane with efficient antibacterial activity and that it could sustain drug release in vitro to achieve longer drug concentration maintenance.

In vitro proliferation, migration, cementoblast differentiation, and in vivo periodontal tissue regeneration were studied by Choung et al. [45] to determine the biological effects of the protein copine7 (Cpne7). In vitro, the effects of rCPNE7 were favorable. The in vivo experiment was conducted on eight beagle dogs, divided into five groups: Group 1—no treatment (negative control); Group 2—collagen carrier only; Group 3—preameloblast-conditioned medium (PA-CM) with collagen carrier (positive control); Group 4—PA-CM + CPNE7 Ab with collagen carrier; and Group 5—rCPNE7 with collagen carrier. From each subject, a mandibular fourth premolar (PM4) were extracted with as little trauma as possible. After the healing period of three months, an enveloped flap was created, extending from the third premolar (PM3) to the first molar (M1) and creating 3 mm × 4 mm × 4 mm periodontal defects. Atelocollagen sheet with a size of 4 mm × 10 mm was soaked in EP tube containing 100 μL of PA-CM or CPNE7 for 5 min, and placed in the bone defect. Groups 1 and 2 showed generation of connective tissue, but no formation of cementum and alveolar bone along the root surface. The PA-CM group regenerated the cementum-PDL-alveolar bone complex similar to the natural periodontal structures. The PACM + CPNE7 Ab group showed the generation of newly formed cementum, PDL, and alveolar bone-like tissue complex, but the fibers in the newly created PDL-like tissue were arranged irregularly. The cementum-PDL-alveolar bone complex was regenerated in the rCPNE7 group, and the newly formed PDL-like fibers were arranged perpendicular to the newly formed cementum and alveolar bone. The fibers were vertically embedded into the newly formed mineralized matrix, such as Sharpey’s fibers, showing the polarity of PDL cells. The periodontal cell’s adhesion to the cementum and the PDL fibers’ physiological organization may be supported by cpne7.

## 4. Discussion

Given that P-D is the leading cause of tooth loss in adults globally, evaluating therapeutic strategies is of paramount importance. The current state of knowledge in the field of P-D treatment consists of the following: systemic antibiotherapy added to open flap debridement (mechanically diminishing the subgingival bacterial biofilm) [13], the use of barrier membranes and bone grafts [46], or the use of guided tissue regeneration strategies [47].

Mechanical debridement, including scaling and root planning, consists in removing plaque, calculus, cementum and dentine contaminated by microorganisms. Periodontitis is a disease with bacterial etiology, and periodontal pathogens have the capacity to grow and form biofilms, which are highly resistant. One of the main advantages of mechanical debridement is that supragingival and subgingival instrumentation can disintegrate bacterial biofilm and reduce bacterial load in periodontal pockets. Periodontal debridement helps in decreasing probing depths and reducing bleeding on probing. Additionally, clinical attachment levels are improved, and bacterial profiles are modified [48]. The main disadvantage of the mechanical debridement is the fact that it leaves a significant number of pathogens in the gingival sulcus, due to the impossible instrumentation of certain areas or due to the ability of microorganisms to penetrate into deeper tissues. Inaccessibility and re-colonization of pathogens can occur after scaling and root planning.

In severe forms of periodontitis, mechanical debridement is less effective and is combined with antibiotherapy to increase the efficiency [49]. Systemic antibiotics enter periodontal pockets by diffusion from the general circulation and can affect profound microorganisms, untreated by local treatments methods. Another advantage of the general administration of antibiotics is that the administration is easy for the patients. Systemic antibiotherapy also affects periodontopathogens located in other areas than periodontal pockets, such as the tongue, tonsils, oral mucosa and other oral surfaces, reducing the potential of recolonization inside the pockets [50]. On the other hand, systemic antibiotics needed for the treatment of periodontal disease have to be administrated in high doses, because the concentration that reaches the periodontal tissues after systemic ingestion is low. Antibiotic resistance is a worldwide issue and antibiotic overuse to treat periodontal disease has definitely contributed. These disadvantages could be avoided with the use of locally applied antimicrobials.

Local antibiotherapy administration is indicated because of its reduced delivery time, high drug concentration at the infection site, and because it eliminates the systemic side effects [51]. Antibacterial biomaterials are of great interest in the innovative treatments of P-D, representing the broadest group of anti-infective biomaterials. As gingival curettage is not associated with regeneration of periodontal structures and bone grafts restore only affected bone, without any effect on the other periodontal structures [52], new therapies are required. Different regenerative treatments have shown the ability to regenerate the periodontal attachment apparatus, although it is extremely expensive, and the outcome is not always predictable [53]. Among these regenerative technologies, guided tissue regeneration (GTR), based on the concept of tissue compartmentalization and homing, has demonstrated the achievement of periodontal regenerative outcomes, although these were often compromised by bacterial colonization and infection, mainly when the barrier membranes were exposed [14]. This treatment approach aims to stop epithelial migration into the area undergoing regeneration by using barrier membranes and bone substitutes. Positive clinical results are produced as a result of the slower migrating periodontal ligament (PDL) cells being able to repopulate the protected area [54], but it has the disadvantage of lacking antimicrobial reduction [14].

A new generation of barrier membranes with enhanced properties is under development. Mucoadhesivity, biocompatibility, non-toxicity, biodegradability, and controlled drug release for a long period of time are prerequisites for localized drug delivery systems. Both membranes and films can satisfy all of these requirements. Considering all the above, the membranes or films loaded with antibiotics or other drugs efficient in periodontal disease, should be taken into account as a better method of treatment for periodontal pockets. In this regard, this review assessed the outcomes of loaded films and membranes in the treatment of P-D, as previously described strategies present limitations.

The first study selected for this review used doxycycline hydroxypropyl methylcellulose membranes to evaluate its effect on the Gram-negative bacteria found in the periodontal plaque of 19 patients. The drug-carrier combination had good results, consisting in significant reduction of periodontal probing depth and of bleeding on probing. These results support the outcome of other clinical studies in literature [55,56,57].

Gamal and Kumper 2012 [35] used EDTA root surface etching as a delivery system of DOX. DOX was loaded on collagen membranes and introduced into the periodontal pockets of human patients. The combination of collagen membranes releasing DOX and EDTA etching proved to be an efficient antibacterial system. Gamal and Kumper emphasized the concept of prolonged drug release using a method that does not interfere with bone regeneration.

Khan et al. [36] conducted a human study in which they tested chitosan films loaded with MZ and LF as a drug release mechanism. The patients treated with chitosan films loaded with both MZ and LF had the best outcome regarding reattachment of the periodontal ligaments, the reduction of periodontal pocket depth, and gingival tissue health. The use of two antibiotics boosted their potential and the authors stated that, besides the beneficial effects, it is a cheap method of managing periodontitis.

In their study, Kassem et al. [37] aimed to develop two types of chitosan films (chitosan–alginate and chitosan–pectin polyelectrolyte complex films), loaded with Tetracycline hydrochloride, in order to evaluate them as drug release systems. The effects of introducing the loaded films into the periodontal pockets consisted of healthy gums with no inflammation, no bleeding on probing, and the reduction of periodontal pockets depth.

Khajuria carried out two studies in vivo on a rat model using chitosan-based films, which they introduced into the periodontal pockets of the animal subjects to inhibit periodontal bone loss. In their first experiment, in 2017 [40], they used chitosan-based risedronate/zinc-hydroxyapatite films, using two different concentrations of risedronate and HA. The results of the study indicated that the intrapocket dental films inhibited periodontal bone loss.

The second study by Khajuria et al., in 2018 [39], presented a chitosan–metformin film tested on animal subjects which had good results in inhibiting bacterial growth and regenerating alveolar bone in the periodontal pockets in which they were applied. Both Khajuria’s studies present innovative methods of limiting the evolution of periodontal disease and even recovering alveolar bone.

Ma et al. [41] studied the effect of poly (lactic-co-glycolic acid) membrane loaded with Minocycline. MINO-PLGA. MINO-PLGA and Perioclin-loaded membrane showed an increase in the height of the alveolar bone. The authors proposed a single administration of the drug-loaded membrane into the periodontal pocket, in an attempt to reduce the patient’s visits to the dental office but not compromise the benefits of the drug effect.

Ho et al. [42] Demonstrated that PDLLA-AMX nanofibres can decrease postoperative inflammation and promote early periodontal repair.

The research group of Kida et al., 2019 [58], tested porous matrices consisting of gelatin and cellulose derivates loaded with metronidazole. The matrices were tested in vitro for physiochemical properties, as well as for cytotoxicity. The clinical study consisted of 23 patients which were divided into two groups (test group and control group). The periodontal tissue degradation was assessed and the metronidazole in polymer matrix was applied to the test group. The periodontal pockets depth was decreased as well as the bleeding, when compared to control group.

In the other three experiments, Wu, Li, Chung [43,44,45] and their research teams used independent drug release carriers to facilitate periodontal regeneration, obtaining satisfactory results.

Natural polymers from animals, plants, algae, and microbes consisting of polysaccharides, polypeptides, and polynucleotides have also been described in the production of membranes with application in periodontal disease [59]. Hydroxyapatite could improve the wound healing in periodontal tissue, while collagen membranes are useful in guided bone regeneration. Furthermore, cathepsin K inhibitors (Ctsk-inhibitors) can stop the destructive process found in the periodontitis. In addition, natural polymers have antifungal properties which could treat Candida albicans, the most common infection of prosthodontic patients [59]. These natural polymer membranes can be carriers for pharmacological agents with a positive effect in the treatment of periodontitis, such as: chlorhexidine, metronidazolem levofloxacin, clindamycin, atorvastatin, moxifloxacin hydrochloride, aceclofenac, or curcumin [59].

New advances in the treatment of periodontits are represented by the personalized therapy, as described by Wirth et al. [60]. A study on 121 patients with periodontitis showed that all biofilms showed three distinct microbial clusters [60]. The most abundant of them was considered to be Tannerella forsythia, and the authors concluded that it is a suitable indicator for periodontitis biofilms [60]. Nevertheless, as different bacterial clusters can be present in the same person, makes the personalized therapy more com-plicated [60].

Between the matrices used as drug carriers in the treatment of P-D could also be included Platelet Rich-Fibrin (PRF). PRF is a three-dimensional scaffold obtained in the dental office from the patient’s blood [61]. Besides its possible role as a carrier, PRF also has many positive effects on wound healing and tissue regeneration, as angiogenesis, gradual growth factor release [62], [63], immunological and antibacterial [64] and even pain release properties [65]. Furthermore, due to its’ autologous origin, it does not activate the organism’s foreign body reaction and it stimulates the body’s natural healing process [66]. All these benefits recommended PRF for various curative applications, including using it as a drug carrier [67]. There are plenty of studies in the literature that tested PRF as a carrier for ampicillin, sulbactam [68], Vancomycin hydrochloride [69] and other drugs. It was used in liquid [70] form as well as in the form of fibrin clot, as a carrier for multipotent cells [71]. The majority of the mentioned studies support PRF as a drug carrier and as a helpful tool in wound healing and tissue regeneration.

The studies using PRF as a drug carrier were not included in this review because the technique for obtaining it is very different compared to artificial membranes and films. The variety of studies available could make PRF a literature review topic on its own [72,73,74,75].

The risk of bias for the included studies was assessed independently by two authors, with the NOS Scale Tool. The assessment process discrepancies were resolved by discussion and reevaluation of the paper. Low risk of bias (7–9 NOS scores), high risk of bias (4–6 NOS scores), and very high risk of bias are the three risk classes that the NOS Scale splits into (0–3 NOS scores). The majority of the research reviewed in this analysis has a low bias risk.

The variability of the studies found in the literature is the primary limitation of this comprehensive review. The publications use various study designs, periodontal disease induction techniques, and medication loaded on membranes and films. Another limitation of this review that could not be avoided is the fact that there are various types of matrices, biomatrices and scaffolds which could be used as drug carriers, but it would have been impossible to conduct an exhaustive search.

## 5. Conclusions

There is a significant public health need for effective strategies for periodontal disease treatment. The innovative treatments should be easily translated into everyday use in clinical dental practice.

The limitations of the current approaches in treating periodontal disease are proved by their limited long-term outcomes, with frequent relapses, thus needing long-term systemic antibiotherapy and periodic curettage.

The use of these controlled drug release devices for intrapocket delivery may herald a new era in the treatment of periodontitis, as they allow for the loading of drugs that specifically target the microorganisms that are causing the pockets to proliferate. This treatment method should be used in localized periodontitis or in cases of patients with a medical contraindication for systemic administration of drugs. In the near future, a new combined approach of local treatments could be applied. Local drug delivery systems could be used in combination with periodontal endoscopy, laser therapy, surgical periodontal therapy in order to reduce the local inflammation, increase tissue regeneration and prevent relapses.

The use of films and membranes loaded with drugs with antimicrobial and regenerative potential appears an appealing alternative to the classical approach to treating periodontal disease, but future research is needed to determine which combination of drug and carrier would be the most effective and to establish a suitable protocol for their use.

## Figures and Tables

**Figure 1 membranes-12-00895-f001:**
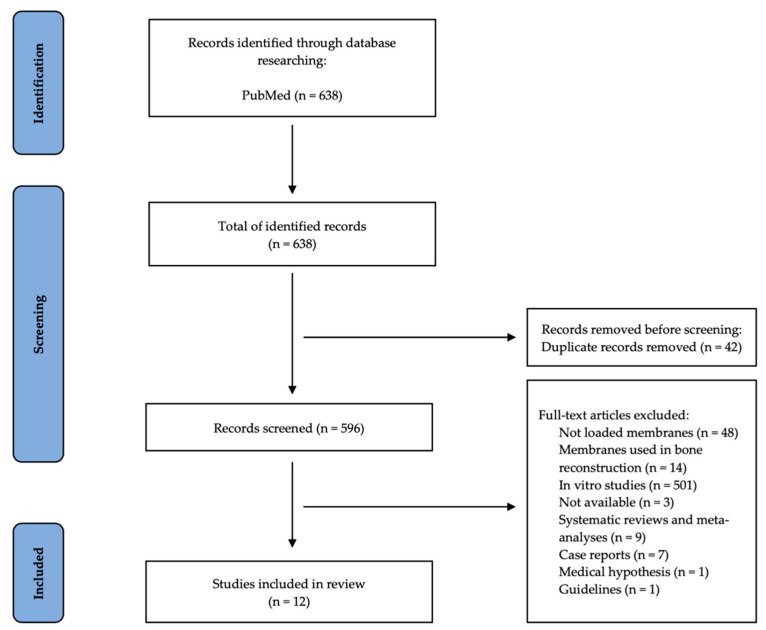
PRISMA Flow Diagram—selection of the included studies.

**Table 1 membranes-12-00895-t001:** NOS Scale of the included articles.

Study	Selection	Comparability	Outcome	NOS Score
Human Studies				
Mahajania et al., 2018 [34]	***	**	***	8
Gamal and Kumper, 2012 [35]	**	**	***	7
Khan et al., 2015 [36]	***	**	***	8
Kassem et al., 2015 [37]	**	**	***	7
Animal Studies				
Liu et al., 2020 [38]	**	*	***	6
Khajuria et al., 2018 [39]	***	**	***	8
Khajuria et al., 2017 [40]	***	**	***	8
Ma et al., 2020 [41]	***	**	***	8
Ho et al., 2021 [42]	***	*	***	7
Wu et al., 2021 [43]	*	**	***	7
Li et al., 2019 [44]	***	**	***	8
Choung et al., 2019 [45]	**	**	***	7

**Table 2 membranes-12-00895-t002:** Human studies included in the review.

References	Number of Patients	Type of Membrane	Loaded Drug	Results
Mahajania et al., 2018 [34]	*n* = 19	Hydroxy-propyl methylcellulose films.	Doxycicline (DOX)	Local release of doxycicline is an effective antibiotic of choice in periodontitis.
Gamal and Kumper, 2012 [35]	*n* = 30	Biodegradable collagen membrane (COL)	Doxycycline (DOX)	The results of the study suggest that the DOX–COL + EDTA-guided tissue antibacterial regimen is a convenient method for obtaining a prolonged drug release without compromising the space that should be occupied by regenerating tissues or by inducing smearing of the root surface by DOX gel.
Khan et al., 2015 [36]	*n* = 10	Biodegradable films of chitosan (CS)	Metronidazole (MZ) and levofloxacin (LF)	The films of MZ and LF were successful for the management of periodontitis.
Kassem et al., 2015 [37]	*n* = 5	Chitosan-alginate and chitosan-pectin polyelectrolyte complex (PEC) films	Tetracycline hydrochloride (Tc)	PEC films could be exploited as a prolonged drug release devices for treatment of periodontal pockets.

**Table 3 membranes-12-00895-t003:** Animal studies included in the review.

References	Type of Subjects (Number)	Type of Membrane	Type of Load	Results
Liu et al., 2020 [38]	Dogs (*n* = 9)	Core-shell nanofiber membrane.	SP600125 and BMP-2	Membranes with sequential release of SP600125 and BMP-2 represent a good therapy for periodontitis.
Khajuria et al., 2018 [39]	Rats (*n* = 40)	Bioabsorbable chitosan-metformin based intrapocket dental film (CMIDF)	Metformin hydrochloride	This study indicates potential antibacterial and osteoprotective efficacy of novel CMIDF in experimental periodontitis.
Khajuria et al., 2017 [40]	Rats (*n* = 60)	Bioresorbable chitosan-based risedronate/zinc-hydroxyapatite intrapocket dental film (CRZHDF)	Risedronate/zinc-hydroxyapatite nanoparticles	The study reported here reveals that novel CRZHDF treatment effectively reduced alveolar bone destruction and contributes to periodontal healing in a rat model of experimental periodontitis.
Ma et al., 2020 [41]	Rats (*n* = 25)	Poly (lactic-co-glycolic acid) electrospun membrane (PLGA)	Minocycline	This study states that Minocycline-loaded PLGA could be used to stimulate bone regeneration in periodontal disease.
Ho et al., 2021 [42]	Rats (*n* = 27)	Poly-DL-lactic acid nanofibers (PDLLA)	Amoxicilin (AMX)	PDDLA-AMX decreased inflammation and accelerated periodontal regeneration.
Wu et al., 2021 [43]	Rats (-)	chitin hydrogel membrane (ChT-1%ZnO)	zinc oxide nanoparticles	The results suggested that ChT-1%ZnO performs better than pure ChT in periodontal bone defects regeneration, but it could not fully repair the defect to the primary level.
Li et al., 2019 [44]	Rats (*n* = 12)	chitosan membrane	polyphosphoester and minocycline hydrochloride (PPEM)	Incorporation of 0.6% (W/V) MH drug provided the membrane with efficient antibacterial activity and that it could sustain drug release in vitro to achieve longer drug concentration maintenance.
Choung et al., 2019 [45]	Dogs (*n* = 8)	collagen carrier	protein copine7 (Cpne7)	The PDL fibers’ physiological organization and periodontal cells’ attachment to cementum may be supported by Cpne7.

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
