# Peer review of "Gradual Drug Release Membranes and Films Used for the Treatment of Periodontal Disease"

_membranes, 2022, doi:10.3390/membranes12090895_

Round 1
Reviewer 1 Report
This paper summarizes the treatment of periodontitis with film loaded drugs. The logic of this paper is clear. However, certain concerns need to be noticed and clarified:
1. The arrow in the Figure 1 is not aligned with the text box, and there are many blank parts in the text box. It is recommended to modify.
2. In the frontier, there are few ways to treat periodontitis with biofilm. The research collection of this article in the last five years is less.
3. This article rarely compares the advantages of other fields in the treatment of periodontitis.
4. The developement direction of this treatment method should be discussed.
Author Response
Dear Reviewer,
Thank you for allowing us to resubmit a revised version of our manuscript. Please accept our revised version for further consideration.
We would like to express our gratitude for providing constructive feedback by identifying the areas of our manuscript that needed further improvements. We appreciate the tremendous effort and time you have devoted to strengthening our manuscript.
Accordingly, we have uploaded the revised manuscript with all the changes indicated with red and the responses to your feedback indicated in blue. Please find below our response to your comments.
We hope this will make the paper easier to read and we are confident that the new version of the manuscript is significantly improved. Thus, we look forward to hearing from you and to respond to any other questions or comments you may have.
With my best regards,
Prof. Dr. Ondine Patricia Lucaciu.
(on behalf of all coauthors)
This paper summarizes the treatment of periodontitis with film loaded drugs. The logic of this paper is clear. However, certain concerns need to be noticed and clarified:
- The arrow in the Figure 1 is not aligned with the text box, and there are many blank parts in the text box. It is recommended to modify.
We redid Figure 1. Your feedback was very useful. Thank you.
Figure 1. PRISMA Flow Diagram – selection of the included studies
- In the frontier, there are few ways to treat periodontitis with biofilm. The research collection of this article in the last five years is less.
Thank you for your observation. We added:
New advances in the treatment of periodontits are represented by the personalized therapy, as described by Wirth et al. [60]. A study on 121 patients with periodontitis showed that all biofilms showed three distinct microbial clusters [60]. The most abundant of them was considered to be Tannerella forsythia, and the authors concluded that it is a suitable indicator for periodontitis biofilms [60]. Nevertheless, as different bacterial clusters can be present in the same person, makes the personalized therapy more complicated [60].
- Wirth, R.; Pap, B.; Maróti, G.; Vályi, P.; Komlósi, L.; Barta, N.; Strang, O.; Minárovits, J.; Kovács, K.L. Toward Personalized Oral Diagnosis: Distinct Microbiome Clusters in Periodontitis Biofilms. Front Cell Infect Microbiol 2021, 11, 747814.
- This article rarely compares the advantages of other fields in the treatment of periodontitis.
Thank you for the remark. We discussed the advantages of other fields in the treatment of periodontitis:
Mechanical debridement, including scaling and root planning, consists in removing plaque, calculus, cementum and dentine contaminated by microorganisms. Periodontitis is a disease with bacterial etiology, periodontal pathogens, having the capacity of growing and forming biofilms, which are highly resistant. One of the main advantages of mechanical debridement is that supragingival and subgingival instrumentation can disintegrate bacterial biofilm and reduce bacterial load in periodontal pockets. Periodontal debridement helps in decreasing probing depths and reducing bleeding on probing. Also, clinical attachment levels are improved, and bacterial profiles are modified [48].
In severe forms of periodontitis, mechanical debridement is less effective and is combined with antibiotherapy to increase the efficiency. [49] Systemic antibiotics enter periodontal pockets by diffusion from the general circulation and can affect profound microorganisms, untreated by local treatments methods. Another advantage of general administration of antibiotics is that the administration is easy for the patients. Systemic antibiotherapy also affects periodontopathogens located in other areas than periodontal pockets, such as tongue, tonsils, oral mucosa and other oral surfaces, reducing the potential of recoloniosation inside the pockets. [50]
- Satheesh, K. Successful Strategies for Periodontal Debridement, Dimensions of Dental Hygiene. 2017, 15, 39-44.
- Guzeldemir-Akcakanat, E. Systemic antibiotics in the treatment of periodontitis. Dent Med Res 2019, 7, 33-4.
- Bogdanovska, L.; Kukeska, S.; Popovska, M.; Petkovska, R.; Goracinova, K. Therapeutic strategies in the treatment of perio-dontitis. Macedonian pharmaceutical bulletin. 2012, 58, 3-14.
- The developement direction of this treatment method should be discussed.
Thank you for your suggestion. We added the perspectives of this treatment method:
This treatment method should be used in localized periodontitis or in cases of patients with a medical contraindication for systemic administration of drugs. In the near future a new combined approach of local treatments could be applied. Local drug delivery systems could be used in combination with periodontal endoscopy, laser therapy, surgical periodontal therapy in order to reduce the local inflammation, increase tissue regeneration and prevent relapses.

Reviewer 2 Report
Dear Authors,
thank you for an opportunity to review this paper. It has some flaws though, so here are suggestions of mine to correct it:
1. in the materials and methods section, please add the weather or not metaanalyses and systematic reviews were excluded
2. I think it would be valid to add this paper to the ones discussed by Authors
- Kida, D.; Karolewicz, B.; Junka, A.; Sender-Janeczek, A.; DuÅ›, I.; Marciniak, D.; Szulc, M. Metronidazole-Loaded Porous Matrices for Local Periodontitis Treatment: In Vitro Evaluation and In Vivo Pilot Study. Appl. Sci. 2019, 9, 4545. https://doi.org/10.3390/app9214545
3. Please, add perspectives in your research. I think you should discuss it with the recent advances in use of natural polymers as a valid tool for matrices, eg.:
- Paradowska-Stolarz A, Wieckiewicz M, Owczarek A, Wezgowiec J. Natural Polymers for the Maintenance of Oral Health: Review of Recent Advances and Perspectives. Int J Mol Sci. 2021 Sep 25;22(19):10337. doi: 10.3390/ijms221910337. PMID: 34638678; PMCID: PMC8508910.
4. It would be valid to add PRP/PRF use as a tool for bone and tissue regeneration in the aspect of matrices, eg. doi: 10.17219/dmp/117721 (there are planty of the articles concerning that)
To sum up, the article is well written and after those small corrections, could be accepted for publishiung
5. Please, add the limitations section (eg. inclusion cryterias, lack of "matrices" in the searched studies)
Author Response
Dear Reviewer,
Thank you for allowing us to resubmit a revised version of our manuscript. Please accept our revised version for further consideration.
We would like to express our gratitude for providing constructive feedback by identifying the areas of our manuscript that needed further improvements. We appreciate the tremendous effort and time you have devoted to strengthening our manuscript.
Accordingly, we have uploaded the revised manuscript with all the changes indicated with red and the responses to your feedback indicated in blue. Please find below our response to your comments.
We hope this will make the paper easier to read and we are confident that the new version of the manuscript is significantly improved. Thus, we look forward to hearing from you and to respond to any other questions or comments you may have.
With my best regards,
Prof. Dr. Ondine Patricia Lucaciu.
(on behalf of all coauthors)
Dear Authors,
thank you for an opportunity to review this paper. It has some flaws though, so here are suggestions of mine to correct it:
- in the materials and methods section, please add the weather or not metaanalyses and systematic reviews were excluded
Thank you for your suggestion. In the materials and methods section this was already mentioned (R141) and we also added the information in Figure 1 (the schematic reprezentation of the article selection process).
Figure 1. PRISMA Flow Diagram – selection of the included studies
- I think it would be valid to add this paper to the ones discussed by Authors
- Kida, D.; Karolewicz, B.; Junka, A.; Sender-Janeczek, A.; DuÅ›, I.; Marciniak, D.; Szulc, M. Metronidazole-Loaded Porous Matrices for Local Periodontitis Treatment: In Vitro Evaluation and In Vivo Pilot Study. Appl. Sci. 2019, 9, 4545. https://doi.org/10.3390/app9214545
Your feedback was very useful. Thank you. We discussed the suggested paper:
The research group of Kida et. al 2019 [58] tested porous matrices consisting of gelatin and cellulose derivates loaded with metronidazole. The matrices were tested in vitro for physiochemical properties, as well as for cytotoxicity. The clinical study consisted in twenty-three patients which were divided into two groups (test group and control group). The periodontal tissue degradation was assessed and the metronidazole in polymer matrix was applied to the test group. The periodontal pockets depth was decreased and also the bleeding, when compared to control group.
- Kida, D.; Karolewicz, B.; Junka, A.; Sender-Janeczek, A.; DuÅ›, I.; Marciniak, D.; Szulc, M. Metronidazole-Loaded Porous Ma-trices for Local Periodontitis Treatment: In Vitro Evaluation and In Vivo Pilot. Study Appl Sci 2019, 9, 4545.
- Please, add perspectives in your research. I think you should discuss it with the recent advances in use of natural polymers as a valid tool for matrices, eg.:
- Paradowska-Stolarz A, Wieckiewicz M, Owczarek A, Wezgowiec J. Natural Polymers for the Maintenance of Oral Health: Review of Recent Advances and Perspectives. Int J Mol Sci. 2021 Sep 25;22(19):10337. doi: 10.3390/ijms221910337. PMID: 34638678; PMCID: PMC8508910.
Thank you for your observation. We added:
Natural polymers from animals, plants, algal and microbial consisting in polysaccharides, polypeptides and polynucleotides have also been described in the production of mem-branes with application in periodontal disease [59]. Hydroxyapatite could improve the wound healing in periodontal tissue, while collagen membranes are useful in guided bone regeneration. Also, cathepsin K inhibitors (Ctsk-inhibitors) can stop the destructive process found in the periodontitis. In addition, natural polymers have antifungal properties which could treat Candida albicans, the most common infection of prosthodontic patients [59]. These natural polymer membranes can be carriers for pharmacological agents with a positive effect in the treatment of periodontitis, such as: chlorhexidine, metronidazolem levofloxacin, clindamycin, atorvastatin, moxifloxacin hydrochloride, aceclofenac or curcumin [59].
- Paradowska-Stolarz, A.; Wieckiewicz, M.; Owczarek, A.; Wezgowiec, J. Natural Polymers for the Maintenance of Oral Health: Review of Recent Advances and Perspectives. Int J Mol Sci 2021, 22(19), 10337.
- It would be valid to add PRP/PRF use as a tool for bone and tissue regeneration in the aspect of matrices, eg. doi: 10.17219/dmp/117721 (there are planty of the articles concerning that)
To sum up, the article is well written and after those small corrections, could be accepted for publishiung
Thank you for your observation. We added:
Between the matrices used as drug carriers in the treatment of P-D could also be included Platelet Rich-Fibrin (PRF). PRF is a three-dimensional scaffold obtained in the dental office from the patient’s blood [61]. Besides its possible role as a carrier, PRF has also many positive effects on wound healing and tissue regeneration, as angiogenesis, gradual growth factor release [62], [63], immunological and antibacterial [64] and even pain re-lease properties [65]. Furthermore, due to its’ autologous origin, it does not activate the organism’s foreign body reaction and it stimulates the body’s natural healing process [66]. All these benefits recommended PRF for various curative applications, including using it as a drug carrier [67]. There are plenty of studies in the literature that tested PRF as a carrier for ampicillin, sulbactam [68], Vancomycin hydrochloride [69] and other drugs. It was used in liquid [70] form as well as in the form of fibrin clot, as a carrier for multipotent cells [71]. The majority of the mentioned studies support PRF as a drug carrier and as a helpful tool in wound healing and tissue regeneration.
The studies using PRF as a drug carrier were not included in this review because its obtaining technique is very different, compared to artificial membranes and films and the variety of studies available could make PRF a literature review topic by its own [72-75].
- Miron, R.J.; Zucchelli, G.; Pikos, M.A.; Salama, M.; Lee, S.; Guillemette, V.; Fujioka-Kobayashi, M.; Bishara, M.; Zhang, Y.; Wang, H.L.; Chandad, F.; Nacopoulos, C.; Simonpieri, A.; Aalam, A.A.; Felice, P.; Sammartino, G.; Ghanaati, S.; Hernandez, M.A.; Choukroun, J. Use of platelet-rich fibrin in regenerative dentistry: a systematic review. Clin Oral Investig 2017, 21(6), 1913–1927.
- Choukroun, J.; Diss, A.; Simonpieri, A.; Girard, M.O.; Schoeffler, C.; Dohan, S.L.; Dohan, A.J.; Mouhyi, J.; Dohan, D.M. Plate-let-rich fibrin (PRF): a second-generation platelet concentrate. Part IV: clinical effects on tissue healing. Oral Surg Oral Med Oral Pathol Oral Radiol Endod 2006, 101(3), e56–e60.
- Egle, K.; Salma, I.; Dubnika, A. From Blood to Regenerative Tissue: How Autologous Platelet-Rich Fibrin Can Be Combined with Other Materials to Ensure Controlled Drug and Growth Factor Release. Int J Mol Sci 2021, 22(21), 11553.
- Melo-Ferraz, A.; Coelho, C.; Miller, P.; Criado, M. B.; Monteiro, M. C. Platelet activation and antimicrobial activity of L-PRF: a preliminary study. Mol Biol Rep 2021, 48(5), 4573–4580.
- Fan, Y.; Perez, K.; Dym, H. Clinical Uses of Platelet-Rich Fibrin in Oral and Maxillofacial Surgery. Dent Clin North Am 2020, 64(2), 291–303.
- Karimi, K.; Rockwell, H. The Benefits of Platelet-Rich Fibrin. Facial Plast Surg Clin North Am 2019, 27(3), 331–340.
- Egle, K.; Skadins, I.; Grava, A.; Micko, L.; Dubniks, V.; Salma, I.; Dubnika, A. Injectable Platelet-Rich Fibrin as a Drug Carrier Increases the Antibacterial Susceptibility of Antibiotic-Clindamycin Phosphate. Int J Mol Sci 2022, 23(13), 7407.
- Straub, A.; Vollmer, A.; Lâm, T.T.; Brands, R.C.; Stapf, M.; Scherf-Clavel, O.; Bittrich, M.; Fuchs, A.; Kübler, A.C.; Hartmann, S. Evaluation of advanced platelet-rich fibrin (PRF) as a bio-carrier for ampicillin/sulbactam. Clin Oral Investig 2022, 10.1007/s00784-022-04663-y. Advance online publication.
- Dubnika, A.; Egle, K.; Skrinda-Melne, M.; Skadins, I.; Rajadas, J.; Salma, I. Development of Vancomycin Delivery Systems Based on Autologous 3D Platelet-Rich Fibrin Matrices for Bone Tissue Engineering. Biomedicines 2021, 9(7), 814.
- Miron, R.J.; Zhang, Y. Autologous liquid platelet rich fibrin: A novel drug delivery system. Acta biomaterialia 2018, 75, 35–51.
- Di Liddo, R.; Bertalot, T.; Borean, A.; Pirola, I.; Argentoni, A.; Schrenk, S.; Cenzi, C.; Capelli, S.; Conconi, M.T.; Parnigotto, P.P. Leucocyte and Platelet-rich Fibrin: a carrier of autologous multipotent cells for regenerative medicine. J Cell Mol Med 2018, 22(3), 1840–1854.
- Rafiee, A.; Memarpour, M.; Taghvamanesh, S.; Karami, F.; Karami, S.; Morowvat, M. H. Drug Delivery Assessment of a Novel Triple Antibiotic-Eluting Injectable Platelet-Rich Fibrin Scaffold: An In Vitro Study. Curr Pharm Biotechnol 2021, 22(3), 380–388.
- Kuehnel, R.U.; Schroeter, F.; Mueller, T.; Ostovar, R.; Albes, J.M. Platelet-Rich Fibrin in Combination with Local Antibiotics Optimizes Wound Healing After Deep Sternal Wound Problems and Prevents Reinfection. Surg Technol Int 2021, 39, 313–316.
- Kornsuthisopon, C.; Pirarat, N.; Osathanon, T.; Kalpravidh, C. Autologous platelet-rich fibrin stimulates canine periodontal regeneration. Sci Rep 2020, 10(1), 1850.
- Bhattacharya, H.S.; Gummaluri, S.S.; Astekar, M.; Gummaluri, R.K. Novel method of determining the periodontal regenerative capacity of T‑PRF and L‑PRF: An immunohistochemical study. Dent Med Probl 2020, 57(2), 137–144.
- Please, add the limitations section (eg. inclusion cryterias, lack of "matrices" in the searched studies)
Thank you very much for your valuable input, as suggested, we added in the discussion section the limitations of this review.
The variability of the studies found in the literature is the primary limitation of this comprehensive review. The publications use various study designs, periodontal disease induction techniques, and medication loadeding on membranes and films. Another limitation of this review, that could not be avoided, is the fact that there are various types of matrices, biomatrices and scaffolds which could be used as drug carriers, but it would have been impossible to do an exhaustive search.

Reviewer 3 Report
Periodontal treatment is a very important topic.
In addition to drugs, there are currently light therapy and debridement modalities.
Suggestions can be added.
In addition, periodontal treatment and regeneration membrane should be separated.
Author Response
Dear Reviewer,
Thank you for allowing us to resubmit a revised version of our manuscript. Please accept our revised version for further consideration.
We would like to express our gratitude for providing constructive feedback by identifying the areas of our manuscript that needed further improvements. We appreciate the tremendous effort and time you have devoted to strengthening our manuscript.
Accordingly, we have uploaded the revised manuscript with all the changes indicated with red and the responses to your feedback indicated in blue. Please find below our response to your comments.
We hope this will make the paper easier to read and we are confident that the new version of the manuscript is significantly improved. Thus, we look forward to hearing from you and to respond to any other questions or comments you may have.
With my best regards,
Prof. Dr. Ondine Patricia Lucaciu.
(on behalf of all coauthors)
Periodontal treatment is a very important topic.
In addition to drugs, there are currently light therapy and debridement modalities.
Suggestions can be added.
In addition, periodontal treatment and regeneration membrane should be separated.
We’ve also introduced periodontal alternative treatment such as periodontal endoscopy, laser therapy and debridement modalities.
As suggested by the reviewer we discussed separetely the periodontal treatment and the use of membranes.
Mechanical debridement, including scaling and root planning, consists in removing plaque, calculus, cementum and dentine contaminated by microorganisms. Periodontitis is a disease with bacterial etiology, periodontal pathogens, having the capacity of growing and forming biofilms, which are highly resistant. One of the main advantages of mechanical debridement is that supragingival and subgingival instrumentation can disintegrate bacterial biofilm and reduce bacterial load in periodontal pockets. Periodontal debridement helps in decreasing probing depths and reducing bleeding on probing. Also, clinical attachment levels are improved, and bacterial profiles are modified [48].
In severe forms of periodontitis, mechanical debridement is less effective and is combined with antibiotherapy to increase the efficiency [49]. Systemic antibiotics enter periodontal pockets by diffusion from the general circulation and can affect profound microorganisms, untreated by local treatments methods. Another advantage of general administration of antibiotics is that the administration is easy for the patients. Systemic antibiotherapy also affects periodontopathogens located in other areas than periodontal pockets, such as tongue, tonsils, oral mucosa and other oral surfaces, reducing the potential of recoloniosation inside the pockets [50].
[...]
This treatment method should be used in localized periodontitis or in cases of patients with a medical contraindication for systemic administration of drugs. In the near future a new combined approach of local treatments could be applied. Local drug delivery systems could be used in combination with periodontal endoscopy, laser therapy, surgical periodontal therapy in order to reduce the local inflammation, increase tissue regeneration and prevent relapses.
- Satheesh, K. Successful Strategies for Periodontal Debridement, Dimensions of Dental Hygiene. 2017, 15, 39-44.
- Guzeldemir-Akcakanat, E. Systemic antibiotics in the treatment of periodontitis. Dent Med Res 2019, 7, 33-4.
- Bogdanovska, L.; Kukeska, S.; Popovska, M.; Petkovska, R.; Goracinova, K. Therapeutic strategies in the treatment of perio-dontitis. Macedonian pharmaceutical bulletin. 2012, 58, 3-14.

Round 2
Reviewer 1 Report
accepted as it is
Reviewer 3 Report
I think the manuscript has been substantially revised.
Suggestions to add references:
Lin, Wei-Chun, et al. "Long-term in vitro degradation behavior and biocompatibility of polycaprolactone/cobalt-substituted hydroxyapatite composite for bone tissue engineering." Dental Materials 35.5 (2019): 751-762.